# Harnessing Liquid Biopsies to Guide Immune Checkpoint Inhibitor Therapy

**DOI:** 10.3390/cancers14071669

**Published:** 2022-03-25

**Authors:** Shadma Fatima, Yafeng Ma, Azadeh Safrachi, Sana Haider, Kevin J. Spring, Fatemeh Vafaee, Kieran F. Scott, Tara L. Roberts, Therese M. Becker, Paul de Souza

**Affiliations:** 1Department of Medical Oncology, Ingham Institute of Applied Medical Research, Liverpool, NSW 2170, Australia; yafeng.ma@unsw.edu.au (Y.M.); sana.haider@health.nsw.gov.au (S.H.); k.spring@westernsydney.edu.au (K.J.S.); kieran.scott@westernsydney.edu.au (K.F.S.); tara.roberts@westernsydney.edu.au (T.L.R.); therese.becker@inghaminstitute.org.au (T.M.B.); p.desouza@westernsydney.edu.au (P.d.S.); 2School of Biotechnology and Biomolecular Sciences, University of New South Wales, Sydney, NSW 2031, Australia; a.safarchi@unsw.edu.au (A.S.); f.vafaee@unsw.edu.au (F.V.); 3School of Medicine, Western Sydney University, Campbelltown, NSW 2560, Australia; 4South Western Sydney Clinical School, UNSW, Sydney, NSW 2031, Australia; 5Centre for Circulating Tumor Cell Diagnosis and Research, Ingham Institute for Applied Medical Research, Liverpool, NSW 2170, Australia; 6UNSW Data Science Hub, University of New South Wales, Sydney, NSW 2031, Australia

**Keywords:** immunotherapy, immune checkpoint inhibitor, biomarkers, liquid biopsy, ctDNA, circulating tumor cells, tumor mutational burden, immune cells, neutrophil to lymphocyte ratio, mutations

## Abstract

**Simple Summary:**

Cancer remains a major cause of morbidity and mortality for millions of people across the globe. While immunotherapy using immune-checkpoint inhibitors (ICIs) is revolutionizing the cancer field, the therapy many a times fails in numerous patients and is accompanied with life threatening side effects. In this review, we have highlighted the necessity for robust and sensitive biomarkers that can identify patients most likely to respond to immunotherapy and further to dynamically monitor treatment effects in a real-time manner. Specifically, we focused on non-invasive liquid biopsy derived circulatory tumour DNA, circulatory tumour cells, and immune cells-based biomarkers. We concluded that emerging efforts will soon help standardise and overcome the associated challenges in the use of liquid biopsy approaches. In the near future these approaches will guide routine clinical decisions for immune therapy.

**Abstract:**

Immunotherapy (IO), involving the use of immune checkpoint inhibition, achieves improved response-rates and significant disease-free survival for some cancer patients. Despite these beneficial effects, there is poor predictability of response and substantial rates of innate or acquired resistance, resulting in heterogeneous responses among patients. In addition, patients can develop life-threatening adverse events, and while these generally occur in patients that also show a tumor response, these outcomes are not always congruent. Therefore, predicting a response to IO is of paramount importance. Traditionally, tumor tissue analysis has been used for this purpose. However, minimally invasive liquid biopsies that monitor changes in blood or other bodily fluid markers are emerging as a promising cost-effective alternative. Traditional biomarkers have limitations mainly due to difficulty in repeatedly obtaining tumor tissue confounded also by the spatial and temporal heterogeneity of tumours. Liquid biopsy has the potential to circumvent tumor heterogeneity and to help identifying patients who may respond to IO, to monitor the treatment dynamically, as well as to unravel the mechanisms of relapse. We present here a review of the current status of molecular markers for the prediction and monitoring of IO response, focusing on the detection of these markers in liquid biopsies. With the emerging improvements in the field of liquid biopsy, this approach has the capacity to identify IO-eligible patients and provide clinically relevant information to assist with their ongoing disease management.

## 1. Background

With over 4000 ongoing clinical trials to identify better therapies for cancer [1] immunotherapy (IO), in particular the use of immune checkpoint inhibitors (ICIs), is rapidly transforming the therapeutic landscape across multiple solid and haematopoietic tumours. Currently, approved checkpoint inhibitors target immune checkpoints such as cytotoxic T-lymphocyte-associated protein 4 (CTLA-4), programmed cell death protein 1 (PD-1), and programmed death-ligand 1 (PD-L1) which function as negative feedback regulators of T-cell function [2,3]. One of the mechanisms by which tumor cells evade immune recognition and T-cell mediated destruction is by expressing PD-L1 on the cell surface. PD-L1 binds to PD-1 expressed on T cells, B cells, or macrophages and inhibits their activity. Antibodies that bind to either CTLA-4, PD-1, or PDL1 can block these “off-switches” and re-program T-cells to attack tumor cells (Figure 1).

Despite encouraging results in multiple cancer types, particularly in melanoma [3], lung cancer [4], and renal cancer [5], approximately 40–60% of patients do not achieve any significant therapeutic benefit from IO [6]. This is perhaps governed by individual germline or cancer specific genetics and further mediated by varying trophic, metabolic, and immunological factors [7]. The identification of biomarkers that enable the prediction of the efficacy of IO are of great interest, given the high costs of these therapies and potential for significant adverse events [5,8,9]. Biomarkers are any measurable biological characteristics that robustly serve as indicators of disease initiation, prognosis and/or treatment response. Currently, tumor tissue-based screening for PD-L1 expression is an FDA approved molecular biomarker utilised in tumor types such as non-small cell lung cancers (NSCLC) or melanoma, where IO is being considered predominantly [8,9,10]. High PD-L1 expression correlates with better response to anti-PD-L1 therapy [7]. Tumor mutational burden (TMB) has also recently been FDA-approved to guide decisions for receiving IO treatment [11,12,13,14].

An index of cancer mutation quantity, mostly a higher TMB, is suggestive of a better response to ICIs [14,15]. Mutations in a cell are normally processed as NEO antigens and presented to T cells by major histocompatibility complex proteins to evoke an immune response against mutation bearing cells. However, tumor cells escape this mechanism by exploiting checkpoint receptors [12,16]. When ICIs re-enable T cell activation, higher TMB results in more neo-antigens, thus increasing the chances for T cell recognition, and clinically correlates with better ICI outcomes [15,17]. However, in some patients with both high TMB and PD-L1 positivity, IO still fails, revealing our incomplete understanding of the factors mediating the response to ICIs [6,7,17].

Often based on a small, resected tumor tissue, the genetic and epigenetic information derived from the biopsy samples are mostly incomplete and unable to reveal the complete intra-tumoral heterogeneity. This induces significant challenges in selecting an effective treatment strategy based only on tissue biopsy [18,19,20]. Furthermore, due to the molecular evolution of tumours over time, tissue biopsies (often only available from the time of initial diagnosis) cannot accurately predict treatment response [21,22]. The invasive nature of tissue biopsies also makes it unreasonable to obtain multiple samples throughout treatment. To overcome these challenges and complement invasive tissue biopsies, minimally invasive liquid biopsy—the sampling of blood or other bodily fluids containing tumour-derived entities analysable for biomarkers—is an attractive alternative [21,23,24]. Liquid biopsy also holds promise to better reflect tumor heterogeneity compared to tissue biopsies since the tumour derived entities might originate from different tumor sites and reflect characterstics from the primary tumor site as well as from metastatic sites. Importantly, capturing these tumor biomarkers in blood provide a tumour snapshot in real time, and when assessed sequentially, it provides a holistic information of the tumor allowing for successive monitoring of the entire tumor heterogeneity and tumor evolution during the course of the treatment [22]. Tumour-derived entities present in blood may include circulating proteins, circulating tumor DNA (ctDNA) and RNA (ctRNA), extracellular vesicles (EVs) and circulating tumor cells (CTCs) [17,23,24] (Figure 2). Recently, circulating immune cells can be used to detect biomarkers corresponding to patient-specific pathological information and function as relevant cancer biomarkers [24,25]. Additionally, the soluble counter parts of PD-1 and PD-L1 (sPD-1 ad sPD-L1) has shown predictive and prognostic value in cancer patients on ICI therapy [26,27] (Figure 2).

Here, we review the current knowledge of these liquid biopsy biomarkers as predictors of response to IO and their potential to guide clinical decisions for IO. We will focus on circulating tumor DNA, circulating tumor cells (CTCs), and the types of markers expressed on peripheral blood mononuclear cells (PBMCs). Other biomarker such as sPD-1, sPD-L1, EVs, and exosomes are also briefly discussed. Table 1 lists the example studies on the prognostic value of ctDNA, CTCs, and immune cells in immunotherapies.

## 2. Circulating Tumor DNA (ctDNA)

Circulating tumor DNA (ctDNA) refers to fragments of DNA shed into the bloodstream from apoptotic and necrotic cancer cells; ctDNA can represent a major genomic reservoir of tumor clones detectable from blood samples [29,52]. For IO, the utility of ctDNA to produce predictive information and its monitoring capacity has been evaluated mostly in prospective studies [31,53]. ctDNA is usually measured by allele frequency or mutant molecules per millilitre. Generally, concordance is observed between ctDNA mutational biomarkers and those detected in tumor tissue for advanced and metastatic cancers [54,55,56]. Multiple IO trials conferred the utility of ctDNA for early diagnosis, prognosis prediction, detecting mutations, identifying minimal residual disease, and monitoring therapy response. Being non-invasive, ctDNA assessments provide a real-time monitoring technique and can be measured throughout the course of treatment to determine a patient’s response to therapy [30] and can help differentiate between pseudoprogression (ctDNA levels are not rising, but the image shows an increase in tumor size) and progression (an increase in tumor size and increased ctDNA levels [54,55]. A decrease in ctDNA levels post initiation of IO usually correlates with increased survival in patients [29]. However, ctDNA detectability varies from 0.01% to more than 90% of total cell free DNA (cfDNA), depending on tumor type, anatomical location, stage of the cancer, as well as the tumor microenvironment. Comparatively low ctDNA levels in patient blood samples with higher background cfDNA unrelated to tumor cells make detection a challenge [53,57]. Therefore, the standard procedure for blood collection and DNA isolation should be carefully selected and the detection technologies need to be extremely sensitive [29,54].

### 2.1. ctDNA Levels as an IO Biomarker

The prognostic and predictive value of ctDNA analysis for IO was demonstrated in multicancer prospective phase II studies where distinct cohorts of patients including high-grade serous ovarian cancer (HGSOC), malignant melanoma, and mixed solid tumours were treated with an anti-PD1 antibody [58,59,60]. ctDNA levels (copies/mL plasma) were assessed at baseline in 94 patients and for every 3 cycles during pembrolizumab treatment in 73 patients where serial plasma samples were available. When interrogated by deep sequencing, a decrease in ctDNA levels after two cycles of pembrolizumab correlated with better patient outcomes irrespective of tumor type [30,60]. Another study assessed baseline and on treatment ctDNA levels of 40 advanced melanoma patients receiving PD-1 inhibitors alone or in combination with ipilimumab (anti-CTLA-4 antibody). Patients with higher baseline ctDNA levels and persistently high ctDNA levels early during treatment had shorter progression free survival (PFS) and overall survival (OS) [58]. Reduction of ctDNA levels post 2–3 weeks of ICI administration was predictive of higher OS [6,56]. Researchers from three recent studies, (1) 1/2 CD-ON-MEDI4736-1108 trial (NCT01693562) [59,61], (2) a phase 2 ATLANTIC trial (NCT02087423) [62], and (3) phase 3 randomized MYSTIC trial (NCT02453282) [63], where the efficacy of a checkpoint inhibitor therapy durvalumab was tested, with or without tremelimumab, confirmed that the pretreatment ctDNA can be employed as a prognostic biomarker (ctDNA detection suggestive of responsive patient), and that on-treatment ctDNA dynamics could predict immunotherapy success (lower levels of ctDNA on treatment). Similarly, in another phase 2 study (NCT02644369) [64,65], where pembrolizumab was given to treat solid tumours, found that a decrease in ctDNA from baseline was linked to immunotherapy benefit. Assessment of ctDNA post first line of therapy can also help in disease monitoring and providing guidance on treatment decisions, for example the results from the IMpower010 phase III study to assess the safety and efficacy of atezolizumab in lung cancer patients in comparison to the best supportive care following resection and adjuvant chemotherapy [66,67]. The next-generation sequencing on ctDNA obtained at baseline and at 9 weeks in a prospective cohort of patients with metastatic NSCLC treated with pembrolizumab-based therapy demonstrated that a reduction in on-treatment ctDNA level is associated with improved response rates at 9 weeks and 6 months, as well as improved progression-free survival and overall survival [66,67]. Similarly, the role of assessing on-treatment changes using ctDNA as a non-invasive means to predict long-term efficacy from pembrolizumab-based therapy in advanced non–small-cell lung cancer (NSCLC) was confirmed by Thompson et al., 2021 [68].

Together, these studies demonstrated that the ctDNA level can be a useful biomarker for assessing the likelihood of response to IO. The change from baseline to early on treatment levels may help inform which patients will benefit from IO and discontinue therapy for patients less likely to benefit at an early stage of therapy when other treatment options, such as chemotherapy or other targeted therapy, may still be of benefit.

### 2.2. ctDNA Mutations as IO Biomarker

Specific somatic alterations detectable in ctDNA may also help in the selection of patients for IO treatment, particularly antiPD-1/anti-PD-L1 agents, across many solid tumours [69]. A panel comprising 710 tumour-associated genes has been used to calculate ctDNA-based mutations derived from 35 melanoma patients treated with ipilimumab and nivolumab (anti-PD-1). The study (NCT02486718) demonstrated an improvement in disease-free survival in patients with stage II to IIIA NSCLC who were treated with adjuvant atezolizumab after surgery, with more evident benefit in patients with PD-L1 expression (HR, 0.66; 95% CI, 0.50–0.88; *p* = 0.004) [66,67]. However, the presence of a phosphatase and tensin homolog (*PTEN*) or a serine/threonine kinase 11 (*STK11*) mutation was correlated with early progression in nonsmall cell lung cancer (NSCLC) patients receiving anti PD-1 IO. In contrast, transversion mutations (substitution of a purine by a pyrimidine or vice versa) in the Ki-ras2 Kirsten rat sarcoma viral oncogene homolog (*KRAS*) gene and tumor suppressor gene *TP53* alone predicted better outcomes [69]. Another study reported that NSCLC patients on anti-PD-1 IO who harbored co-mutations of *STK11* and *KRAS* (*n* = 36) detected in ctDNA had longer OS in comparison to patients who harbored *STK11* mutations alone (13.6 ± 3.4 months, *p* = 0.049, *n* = 37) [70]. The OAK and POPLAR clinical trials showed that mutations in the kelch-like ECH-associated protein 1 (*KEAP1*) and nuclear factor erythroid-2-related factor-2 (*NFE2L2*) genes detected in ctDNA were associated with poorer OS and PFS (OS: HR = 1.7, *p* < 0.001; PFS: HR = 1.4, *p* < 0.001) in NSCLC patients receiving IO [71]. Similarly, another study observed that NSCLC patients who had ctDNA detectable AT-rich interacting domain containing protein 1A gene (*ARID1A*) mutations or AT-rich interacting domain-containing protein 1B gene (*ARID1B*) mutations had a beneficial response to anti PD-(L)1 IO and a prolonged PFS [72].

Some of the mutations commonly detected in ctDNA are *TP53*, *KRAS*, and *BRAF* [73]. In an investigative analysis to develop a mutational signature and predict the response to anti-PD-L1 therapy, a risk model consisting of mutations associated with eight genes (*TP53*, *KRAS*, *STK11*, *EGFR*, *PTPRD*, *KMT2C*, *SMAD4*, and *HGF*) was designed to classify patients into high and low risk groups. Patients with more mutations in their mutation signature had shown longer PFS. The mutation signature was demonstrated as an independent predictive factor for anti-PD-L1 therapy when compared with TMB assayed from tissue [74]. However, this mutation signature is not well correlated with outcomes in wider populations and its specific usefulness is thus uncertain.

## 3. Tumor Mutational Burden

The total number of somatic mutations per DNA megabase (Mb), also known as tumor mutational burden (TMB), is an independent predictor of response to IO across multiple cancer types [75,76]. Biological processes contributing to elevated TMB can result from environmental factors such as exposure to cigarette smoke and ultraviolet radiation, or from deleterious mutations in mismatch repair leading to microsatellite instability, or in the DNA repair machinery [77,78]. In order to determine TMB, a large number of genes must be sequenced, typically more than 300 genes, which are then analysed to determine the number of non-synonymous mutations per mega base pair (Mbp). Association of ctDNA-based TMB i.e., cTMB with clinical outcomes in patients are being explored prospectively in several other studies, including BF1RST study [79], MYSTIC [63], and OAK [80] trials. Overall, patients with detectable ctDNA and higher cTMB at diagnosis (>10–16 mut/Mb) have a longer median OS with at least first line of ICI treatment. For example, in the OAK study, the efficacy of durvalumab plus tremelimumab in first-line NSCLC treatment showed a correlation between cTMB and response to IO. In this study, the benefit of IO was not found in patients with less than 10 mutations per Mb detectable from ctDNA [63]. In parallel, high cTMB is predictive of clinical benefit with IO versus chemotherapy, and the benefit is higher with higher mutations per Mb cutoffs of ≥20 relative to chemotherapy [63,81]. However, in other studies, the concordance of cTMB with tTMB for the enrolled patients was low [82,83]. This discrepancy may be attributed to either variation in the original amount of ctDNA shed by the tumor and normalisation for ct vs. cfDNA may help with the applicability of cTMB. Alternatively, it may be due to technical variations as different ctDNA isolation methods and sequencing methods resulting in different coverage of genomic regions. Broadly, a high TMB is usually defined as having more than at least 10 mutations/Mbp [84], but the TMB determined by targeting panel sequencing (with more than 300 genes) and whole exome sequencing differs greatly, so the TMB score has to be adapted depending on the sequencing method [85,86]. Though tissue biopsy remains the gold standard for TMB evaluation, it is challenging to obtain adequate tissue from advanced cancer patients and the use of archived primary tumor samples may not adequately reflect tumor evolution during progression to the advanced cancer [87]. Thus, a minimally invasive approach by ctDNA-based TMB may be crucial in these cases to identify patients who may benefit from ICI immunotherapies. Some studies show strong concordance of ctDNA-based TMB (cTMB) with tissue TMB (tTMB), suggesting that cTMB testing is feasible and has predictive value for the clinical outcomes of ICIs [52,54,82].

While specific cut-off values for cTMB and tTMB were developed to classify likely responding vs. nonresponding patients, further efforts are needed to fully evaluate the predictive value of cTMB for IO [88].

## 4. Circulating Tumor Cells

### 4.1. CTC Enumeration and IO

Circulating tumor cells (CTCs) are cells shed from primary or metastatic tumor sites into the bloodstream. The number of detectable CTCs in a given volume of blood is a prognostic biomarker in various cancers. Higher CTC counts measured with the FDA approved CellSearch (Janssen Diagnostics) platform correlated with tumor burden and poor patient outcomes in breast, prostate, and colon cancer and CTC counts have been applied as an outcome measure in phase II and III clinical trials [89]. In addition, the reduction in CTC count upon treatment correlates with tumor shrinkage and treatment efficacy [41]. Recent studies have also investigated the prognostic value of CTCs in NSCLC and liver cancer patients treated with IO; the reduction of CTC number was suggestive of response and improved overall survival [89].

### 4.2. CTC Characterisation and IO

With recent technological advances, biomarkers detected in enriched CTC samples or individually analysed CTCs can be assessed [88]. For example, trials targeting the PD-L1/PD1 checkpoint in NSCLC and head and neck cancer have evaluated the predictive power of PDL1 expression on CTCs, however the results have been inconsistent [20,38]. Larger studies are needed to clarify whether PD-L1 detection on CTCs is concordant with tissue PD-L1 expression and more importantly, if it is predictive of ICI response. In melanoma patients, melanoma antigen recognised by T cells 1 (*MART-1*), paired box gene 3 (*PAX3*), and melanoma associated antigen 3 (*MAGE-A3*) mRNA expression in CTCs was evaluated for utility as ICI biomarkers [90] and recently β-catenin (*CTNNB1*) overexpression in CTCs was found to be associated with progressive disease of stage III/IV melanoma patients treated with ICI [91]. CTCs have also been analysed via mRNA sequencing for major histocompatibility complex (MHC) changes, which could potentially be applied to CTCs from ICI clinical trials and tested for correlation with patient outcomes [92].

Due to the rarity of CTCs and the technical difficulties of CTC enumeration and characterisation, the utility of CTC in ICI is still in its early stages, however, there are sufficient data to suggest that larger trials are warranted.

### 4.3. Crosstalk between CTC and Immune Cells

Similar to the ‘do not find me’ signal via PD-L1, CD47 expression on CTCs act as a ‘do not eat me’ signals. A high CD47 expression on tumor or CTC surfaces indicates a strong migration and invasion and renders them protected from phagocytosis from immune cells, such as T cells, NK cells and macrophages [93]. Blocking PD-L1 and CD47 on CTCs increases the production of PD1^+^ B-cells while blocking CD47 on CTCs promotes phagocytosis by macrophages and stimulation of tumour-specific cytotoxic T cells. By blocking both PD-L1 and CD47 on CTCs, the immune system can maintain a higher quality of T cells and natural killer (NK) cells to eliminate tumours [93,94]. A recent study in metastatic breast cancer patients demonstrated that CTCs expressing high CD47 and/or high PD-L1 were associated with faster disease progression and shorter PFS [95]. These markers were independent predictors of increased risk of relapse and death. In addition to supporting a predictive role for CD47 and PD-L1 in these patients, the findings indicate the potential importance of both innate and adaptive immune evasion mechanisms in the metastatic potential of breast cancer in this cohort [93].

## 5. Peripheral Blood Cell-Based Biomarkers

Blockade of the CTLA-4 and PD-L1 pathways through ICIs remodels the pool of circulating immune cells [96]. These changes can be detected through immune profiling of circulating peripheral blood mononuclear cells (PBMCs), including T lymphocytes and NK cells, as well as absolute cell counts of neutrophils, lymphocytes, and myeloid derived suppressor cells. Baseline counts as well as dynamic changes in these cell lineages can provide important insights into the interaction between the host immune system, introduction of ICIs, and cancer. Several studies have explored the role of these cells as early biomarkers of response to ICIs [24,97].

### 5.1. T-cell Receptor Repertoire

T cell receptors (TCRs) recognise tumor specific neo-antigen peptides as foreign to the immune system and subsequently undergo clonal proliferation [98]. T cell receptor diversity is generated by random and imprecise rearrangements of segments of the TCR alpha and TCR beta genes in the thymus and is critical in allowing the adaptive immune system to respond to a variety of pathogens [99]. Peripheral T cell receptor diversity can be assessed by sequencing the CD region 3 of the T-cell receptors and several studies have explored both the baseline T cell repertoire and changes following exposure to IO as possible biomarkers for response to ICIs. Baseline T cell receptor diversity has been hypothesised to predict the response to ICI therapy as an anti-tumor T cell population may be present and become active when ICIs are introduced. Postow et al. [47], in a small series of 12 patients receiving ipilimumab for the treatment of metastatic melanoma explored T cell receptor diversity at baseline and demonstrated significant differences between patients with and without clinical benefit [47]. Similarly, a study conducted by Hogan et al. [48], utilised a PCR-based method to detect clonal expansion of reactive T cells. Interestingly, in this series, despite small numbers, patients with low T cell receptor diversity at baseline were less likely to respond to anti-CTLA-4 therapy, but more likely to respond to anti-PD-L1 therapy, suggesting a role for this index in treatment selection [47]. Changes in the T cell repertoire in response to treatment have also been compared between baseline and post-treatment samples. Robert et al. [100] were able to demonstrate a 30% increase in unique productive sequences of TCR V-beta CDR3 in 19 out of 21 patients, and a median decrease of 30% in only 2 out of 21 patients. However, no significant differences were noted between responders and non-responders [99]. Various confounders have also been recognised to exist, influencing TCR repertoire diversity, such as an age associated decrease in diversity [101], genetic factors such as HLA polymorphisms, and the presence of chronic infections [102].

### 5.2. PD-1 and CD8 Double-Positive T Lymphocytes

PD-1^+^ CD8^+^ T cells represent a sub-population of T cells that are targeted, or rescued, by PD1 blockade. These ‘exhausted’ T cells may become reactivated on initiation of checkpoint inhibitor therapy and early proliferation in this population of cells has been linked with positive clinical outcomes following exposure to an anti-PD-L1 therapy.

Patients with high PD-1^+^ CD8^+^ T-cell receptor diversity at baseline showed better treatment responses and improved PFS, compared with populations that demonstrated low diversity (6.4 months vs. 2.5 months, HR 0.39; *p* = 0.021) [103]. In addition, patients with high PD-1^+^ CD8^+^ TCR clonality after exposure to checkpoint inhibitors also had longer PFS [102]. Similarly, Mazzachi et al. [104] observed a two-fold increase in the percentage of and absolute number of PD-1^+^ CD8^+^ lymphocytes in patients with non-small cell lung cancer who derived clinical benefit, compared to those who did not from the use of nivolumab therapy [104]. Similar findings were reported by Kamphorst et al. [105] who demonstrated an increase in Ki67 ^+^ PD 1^+^ CD8^+^ T cells, indicating conversion to an effector phenotype. Increases were seen within 4 weeks of treatment initiation. In addition, most patients who had disease progression showed either a delayed or absent response in this subpopulation of T cells.

### 5.3. Natural Killer Cells

Although there has been significant focus on the role of T cells, there is emerging evidence for a role for NK cells in mediating an immune response to cancer. NK cells are cytotoxic lymphoid cells and are part of the innate immune system. They work to eliminate malignant cells through the secretion of various immunomodulatory cytokines. They also play an important role in triggering the adaptive immune response.

Based on the relative expression of the surface markers, NK cells can be classified into the CD56^bright^ CD16^dim^ subset and CD56^dim^ CD16^bright^ subset, with the CD56^dim^ CD16^bright^ considered terminally differentiated and exerting direct cytotoxic effector activities through secretion of perforin and granzyme [106].

In the tumor microenvironment PD-1 expression on NK cells is induced by many cancers, including head and neck, thyroid, Hodgkin’s lymphoma and digestive tract tumours resulting in the down regulation of NK cell activity [107]. PD-1^+^ natural killer cells have a weaker anti-tumor function than PD-1^−^ NK cells [107]. Blockade of signal transmission between PD-1 and PD-L1 using checkpoint inhibitors has been proposed to also reactivate the function of NK cells [108]. This role of NK cells is particularly important in tumours where MHC-1 loss on the tumor surface can impact the efficacy of the treatment [105]. NK cells have been investigated as a potential biomarker of response.

Mazzaschi et al. [104] demonstrated that NK cells and CD56^dim^ NK cells carrying receptors NKG2D, NKp30, and NKG2A were nearly 2-fold higher (*p*  <  0.05) in peripheral blood of patients who derived clinical benefit, compared to those who did not [104]. Cho et al. [109] also conducted a smaller study of nine patients, assessing NK cell populations in PBMCs as well as activity. The percentages of NK cells in PBMCs were prominently elevated in the IO responders, compared with non-responders [109].

### 5.4. Myeloid Derived Suppressor Cells

Myeloid derived suppressor cells (MDSCs) are closely related to neutrophils and macrophages. These cells are present at low frequency in healthy individuals but significantly increase in various pathological conditions, such as cancer and chronic inflammation [110]. Myeloid derived suppressor cells cause immune suppression, mainly targeting T cells, both in antigen specific and non-specific ways [111]. The presence of myeloid derived suppressor cells has been linked with worse prognosis in patients with malignancy [111,112,113]. There are two subclasses of MDSC—those more monocytic M-MDSC and polymorphonuclear PMNMDSCs, which express different T-cell surface markers and have some differing functions.

These cells have been proposed as a biomarker of IO response. One study looked at patients with NSCLC receiving nivolumab, following progression on chemotherapy, showing that after the first dose of nivolumab, the proportion of PMN-MDSCs in peripheral blood was significantly higher in non-responders than responders [44]. On the other hand, Passaro et al. reported that patients with higher PMN-MDSCs and a low CD8: PMN-MDSC ratio had significantly improved responses to IO [114].

There are several limitations with the use of MDSCs; MDSC subsets lack uniformly used definitions and a better understanding of the nature of these subsets followed by standardising assays may help to avoid current inconsistencies. Molecular mechanisms governing these cells and molecular markers for recognising them are still being studied [110].

### 5.5. Absolute Lymphocyte Counts and Neutrophil to Lymphocyte Ratio (NLR)

Absolute values and ratios of various blood cells have also been explored in peripheral blood, as biomarkers of response to IO including absolute neutrophil count, absolute lymphocyte count, and neutrophil to lymphocyte ratios.

Robert et al. [100] demonstrated a median increase of 11.1% in absolute lymphocyte count from baseline to days 30–60 in patients receiving IO. Patients who started with a lower baseline count of lymphocytes did not achieve an objective response, whilst responders had a high baseline lymphocyte count.

Several studies have demonstrated high neutrophil to lymphocyte ratio (NLR) as a predictor for poor response to IO. Criscitiello et al. [115] demonstrated that a higher derived neutrophil to lymphocyte ratio was associated with reduced PFS (HR 2.29, *p* = 0.001) and reduced OS (HR 2.06, *p* = 0.02). A meta-analysis of 14 retrospective studies with 1225 non-small cell lung cancer patients [116] further provides evidence for a higher baseline NLR being associated with poor PFS (HR 1.44, *p* < 0.05) and OS (HR 1.75, *p* < 0.05), following treatment with nivolumab. Another metastudy with a total of 17 trials with 2106 lung cancer patients further confirmed that high pretreatment NLR is significantly associated with poorer PFS (HR = 1.44, *p* < 0.001) and OS (HR = 2.86, *p* < 0.001) compared with those with low pretreatment NLR [117] Consistently, the pre- and post-treatment peripheral NLR was negatively correlated with the PFS in multiple cohorts, however, the NLR ratio as a prognostic indicator is not unique to ICI-treated patients as the NLR ratio is prognostic in NSCLC treated with an EGFR inhibitor as well as in other settings. Consequently, NLR may be an indicator of the overall ‘health’ of the immune system and its potential to develop a robust anti-tumor response.

## 6. Other Biomarkers

### 6.1. Soluble PD1 and PD-L1

PD1 and PD-L1 is present in a membrane-bound form in tumor cells and immune cells. However, these may also be secreted in circulation as soluble forms referred to as sPD1 and sPD-L1, and their elevated levels have been generally associated with advanced clinical stages and worse prognosis for cancer patients [28,118,119]. In a study receiving ipilimumab-based treatment for advanced melanoma, researchers found that patients who had a ≥1.5-fold increase in sPD-L1 within 4.5 months post treatment experienced a progressive disease [118]. Similarly, a high pre-treatment sPD-L1 levels were associated with advanced stage in a cohort consisted of 128 patients with non-small cell lung cancer (*n* = 50), melanoma (*n* = 31), small cell lung cancer (*n* = 14), urothelial carcinoma (*n* = 13), and other cancers (*n* = 20). However, the results do not correlate with the tumor PDL1 levels, which was an unanticipated finding, as the levels of sPD-L1 are expected to reflect the expression of tissue PD-L1 [119]. Similarly, a meta-analysis including 1188 advanced lung cancer patients confirmed that high sPD-L1 post treatment was significantly associated with worse OS (HR = 2.20; 95%; *p* < 0.001) and PFS (HR = 2.42; 95% *p* < 0.001) in patients treated with ICIs [26]. Elevated plasma levels of sPD-L1 have displayed a consistent outlook and consistently correlated with worse prognosis in many studies [119]. However, sPD-1 levels have remained unpredictable in their predictive and prognostic ability [1]. Pretherapeutic higher sPD-1 plasma levels have shown to predict advanced disease state and to a lesser extent a worse prognosis. Any increase in sPD-1 plasma level post therapeutically have been correlated with improved survival for various cancers [27,120]. Multiple analysis demonstrates serum PD-1/PD-L1 as an independent predictor of survival upon anti-PD-1 therapy [120]. Serum PD-1 and PD-L1 has the potential to select patients for PD-1-based therapy, however, the reported levels of PD-L1 and PD-1 can be highly variable, due to differences in techniques used to quantify them and their associated limitations. Larger studies and standardisation of the technique should be undertaken to reveal the significance of changes in sPD-L1 and sPD-1 levels for each carcinoma type.

### 6.2. Extracellular Vesicles/Exosomes

Emerging studies are establishing the role of tumour-derived extracellular vesicles (TD-EVs) in immunological cross-talk with the potential to consider a novel biomarker for cancer immunotherapy [121,122]. EVs are lipid enclosed membranes that are released by cells and contain compartments representative of their intracellular origin [9]. EV-based liquid biopsy can potentially identify RNA landscape of tumor (mRNA, miRNAs, and other small RNAs), DNA mutations, tumor specific proteins and expressed neoantigens, without the need for tissue samples [121,123]. Moreover, EVs can monitor T-cell reactivity in patients who are treated with immune checkpoint blockade therapy. In a recent study, Serrati et al. demonstrated that the presence of circulating PD1+ EVs drives resistance to anti-PD1 therapy, and performed a multivariate analysis to show that high level of PD1+ EVs, from T cells and B cells, and high level of PD-L1+ EVs from melanoma cells are both able to predict the response to immunotherapy treatment independently [124]. Similarly, Chen et al. demonstrated the presence of PD-L1 on melanoma derived exosomes (a type of EV) and reported that higher levels of circulating exosomal PD-L1 negatively correlated to a poor clinical outcome after ICI therapy [125]. RNA, protein or DNA analysis from exosomes/EVs may provide biomarkers for ICI response prediction and prognosis.

In another study, Dou et al., 2022, showed that EVs can inhibit CD8+ T-cell activation and cytokine production in vitro in response to T-cell receptor stimulation. Importantly, an anti-PD-L1 blocking antibody significantly reversed the EV-mediated inhibition of CD8+ T-cell activation, suggesting a prognostic values PD-L1 + ve EVs in predicting the effectiveness of therapy detection, as well as a new strategy to enhance T-cell-mediated immunotherapy against SCLC cancers. When tested on another lung cancer cohort, the presence of EVs containing PD-L1 significantly correlated with the progression-free survival of patients on immunotherapy [126]. PD-L1 expressed on the EVs, not the soluble form of PD-L1, is also shown to be associated with disease progression in head and neck cancer [127]. Consistently, in patients with melanoma, TDE PD-L1 is also a marker of immune activation early on after initiation of therapy with PD1-targeting antibodies and predicts a clinical response to PD1 blockade [125,128]. In addition to PD-L1, other immunosuppressive molecules such as, TGFβ1, TRAIL, COX2, CD39/CD73, FASL, TRAIL, and NKG2D were also found to be enriched in TDEs and were able to induce T-cell suppression, suggesting that the investigation of these markers may individually or in combination with PD-L1 can help in predicting the response to the immunotherapy [128,129]. Taken together, the analysis of EV population by liquid biopsy is a promising tool to stratify metastaic patients for immunotherapy.

## 7. Challenges in Liquid Biopsy-Based Biomarker Development

Emerging predictive biomarkers for IO include PD-L1 expression on CTCs, and/or PBMCs, and the assessment of tumour mutational burden [18,20,23,53]. However, the accuracy of predictions of patient response based on these biomarkers is as uncertain as that of tissue-based markers and is often due to technical reasons regarding sensitivity and specificity [12,22,130]. Hence, identifying and overcoming these challenges can pave the way for improving and increasing the reproducibility of these biomarkers.

### 7.1. Intrinsic Limitations of Liquid Biopsies as Biomarkers

Despite the attractiveness of isolating and analysing tumour-associated entities such as ctDNA and CTCs from blood samples, the use of liquid biopsies for prognostic and predictive biomarker detection has yet to significantly influence clinical practice, and testing for these markers using liquid biopsies largely has not yet secured accreditation by leading health authorities such as the FDA and EMA for many [8]. Another important limitation is the scarcity of ctDNA and CTCs in patient blood, especially for some tumour types, in the early stages or anatomical locations that intrinsically shed less ctDNA or CTCs into circulation [130,131]. Technologies have advanced greatly in recent years and the detection sensitivity for ctDNA is at the level of single molecules when measured with droplet digital PCR (ddPCR) [132]. However, simple normal distribution modelling predicts that very low blood concentrations of ctDNA are associated with considerable sampling bias, such that the number of ctDNA molecules (e.g., mutant KRAS) in a blood sample may vary by chance. For instance, if the average number of ctDNA molecules in a 10 mL patient blood sample were 3, the probability to indeed collect exactly 3 molecules in any given 10 mL blood draw is only ~40%, 70% for 3 ± 1 and 95% for 3 ± 2 molecules and in fact the chance of capturing no molecule at all is a possibility, although unlikely (~2%).

These considerations highlight that the interpretation of measurements specifically in the low ctDNA blood concentration range is intrinsically challenging irrespective of assay sensitivity. However, interpretation is also affected by technical parameters such as plasma processing methods, and timing from blood draw, followed by ctDNA extraction and ctDNA detection efficiencies. ctDNA has, overall, been the more reliable marker compared to CTC counts in part due to generally higher molecule numbers per mL blood at least when measured prior to treatment.

For CTCs, sampling bias per blood draw may be more of an issue since numbers per 10 mL blood sample are commonly in the low single digits, even at baseline, unless the patient has advanced disease. Additionally, isolation techniques are more prone to variation due to the complexity of CTC extraction in general. Many depend on cell surface molecule expression, which can be heterogeneous between CTCs and may even evolve with certain therapies; for example, epithelial to mesenchymal transition phenotypes are often associated with therapy resistance and CTCs of a more mesenchymal phenotype reduce or lose EpCAM expression, a critical marker for CTC detection. This makes the use of CTC counts as a biomarker for individual patients challenging, although it still tends to show correlation to outcomes in large patient cohorts [89]. Focusing on CTC-based biomarker expression analysis rather than CTC counts may ultimately prove more beneficial and leverages one of the unique qualities of CTCs, namely being able to screen actual cells of tumour origin for proteins/RNA/DNA.

ctDNA and CTC analysis remain attractive techniques, especially if no up-to-date tissue sample is available, for longitudinal assessments and monitoring therapy response. Increasing the amount of blood per sample as well as monitoring for more than one ctDNA-based marker is helpful in increasing the sensitivity of ctDNA detection. The same is true for CTC counts as well as CTC-based marker detectability [53,89]. Another important step for translation into clinical use is the development of standardised diagnostic methods that ensure comparable data even from different sites and operators for both ctDNA and CTC analysis and agreeing on a common best practice blood collection basics, following strict logistic parameters, and optimising the time window when a sample is processed after collection, to name a few, to get the best possible information from the precious sample.

### 7.2. Challenges with Markers Based on Immune Cells

In numerous studies, the presence of specific T cell subtypes, NK cells and other myeloid cells have been associated with the prediction of response to IO. Several clinical trials have also reported detecting PD-L1, PD-L2, and PD1 expression on either CTCs or circulating immune cells to predict the response to the IO [25,38,126]. However, the data are insufficient to clearly show the predictive power for IO.

In addition to the limited randomised trial and first-line data, the technical variability of the assays used—such as variation in the antibodies used to detect specific markers (e.g., PD-L1)—or the thresholds used to define positivity, compound the challenges discussed. The highly dynamic and heterogeneous nature of immune cells in peripheral blood of cancer patients renders biomarker development exceptionally challenging. Notably, the limitation of many studies is related to small patient cohorts tested and highlights the need to strengthen those findings in more extensive studies [133]. Most of these studies were retrospective rather than prospective, so that analysis of the differences in potential biomarkers between baseline, on treatment and post-treatment samples was not performed in a longitudinal fashion. Thus, more work is needed to enhance the potential utility for immune related biomarkers in circulation.

#### Future Aspects

One major therapeutic breakthrough in medical oncology has been the introduction of IO with checkpoint inhibitors targeting the CTLA-4 and PD-1/PD-L1 axis. However, the issue remains that a significant proportion of patients do not benefit from existing IO therapies, both alone and in combination. To overcome the challenges and offer liquid biopsies as a guide for clinical decisions relating to IO, we not only need a substantial improvement in currently used markers added to PD-L1 expression and TMB, but also require that they have detection methods of sufficient sensitivity, specificity, and predictive power. If sufficient sensitivity, specificity, and predictive power is achieved, these methods are likely to secure accreditation by leading health authorities such as FDA and EMA, which is a prerequisite for moving into a clinical setting.

While a lot of work has focused on the tumour microenvironment and its multifaceted response to IO, measurement of single biomarkers does not adequately capture these complex interactions and tissue biopsy for this purpose is often not available longitudinally or in a timely fashion with respect to therapy commencement. Further “historic” tissue, obtained in some instances years earlier may not adequately represent the biology of the current cancer or the potential mechanisms underpinning treatment resistance to ICIs. A multimodal approach may be used for patient monitoring, including evaluation of radiographs and ctDNA levels, as well as the patient’s quality of life, their performance status, and any reported adverse side effects. Metagenomics and host multi-omics big data integration has been gaining interest in recent years, leading to a rapid evolution of knowledge concerning human–microbiota interactions, new multivariate diagnostic/prognostic biomarkers, and its application to IO therapeutics.

Identification of multimodal biomarkers coupled with an algorithm designed to extract meaningful information from these multiparametric biomarkers followed by in-depth clinical validation using large sample sets is needed. With the current prevalence of IO clinical trials, we expect a prominent increase in validation studies and registration of several blood-based assays as a potential diagnostic, prognostic, monitoring, and therapeutic tool. Such studies will be essential to enable the development of new companion diagnostics able to guide precision medicine in IO.

## Figures and Tables

**Figure 1 cancers-14-01669-f001:**
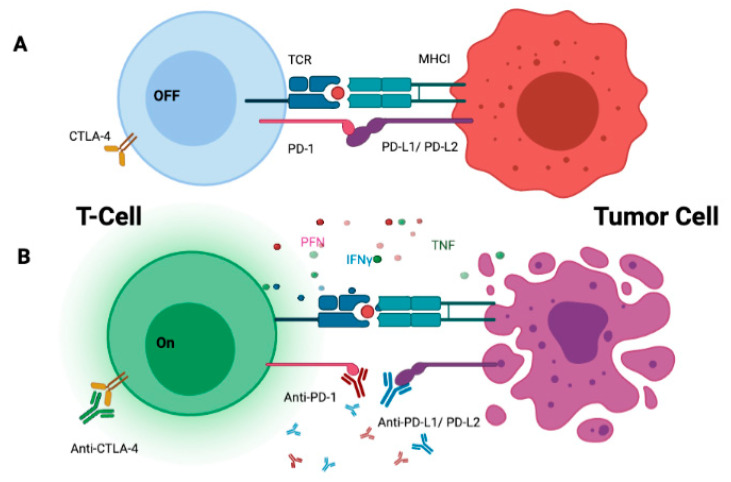
Schematic illustration of mechanism of action of PD-1, PD-L1, and CTLA4 checkpoint inhibitors. (**A**) The programmed cell death 1 (PD-1) receptor is expressed on sensitised immune cells (T cells in representation). Binding of PD-1 to its B7 family of ligands, programmed death ligand 1 (PD-L1) or PD-L2 results in the suppression of antigen-specific T-cell immunologic responses. Similarly, CTLA4 (cytotoxic T-lymphocyte-associated protein 4), is a protein receptor that functions as an immune checkpoint and downregulates immune responses. (**B**) Antibody blockade of PD-1 or PD-L1 or CTLA4 reverses this process and enhances antitumor immune activity. This ultimately leads to the release of cytolytic mediators, such as perforin and granzyme, TNF, and IFNα, causing enhanced tumour killing. TCR, T-cell receptor; MHC, major histocompatibility complex.

**Figure 2 cancers-14-01669-f002:**
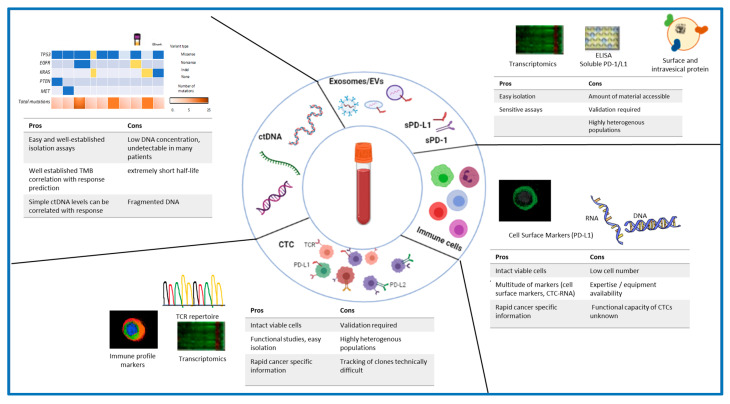
Liquid biopsy analysis in the immunotherapy context. Major sources for immunotherapy biomarkers derived from blood. Sources of liquid biopsy biomarkers are enlisted with major advantages (pros) and disadvantages (cons) associated with their application in clinical settings.

**Table 1 cancers-14-01669-t001:** Representative studies on the prognostic value of ctDNA, CTCs, and immune cells in various immune checkpoint inhibitor therapies.

Cancer Type	Immunotherapeutic Strategies	Measurement Index	Data Outcomes	Reference
**ctDNA**				
**Metastatic NSCLC**	Durvalumab ± tremelimumab	GuardantOMNI ctDNA platform	ctDNA-based TMB (cTMB ≥16 mut/Mb) correlates positively with tissue (t) TMB (≥10 mut/Mb) and is predictive of survival benefit	[28]
**Advanced renal cell carcinoma**	Nivolumab plus ipilimumab	Targeted deep sequencing	Concordance of variants with biopsy-based sequencing tests, increased ctDNA levels post 1 month of treatment showed intrinsic resistance, decreased ctDNA levels showed higher response	[29]
**NSCLC**	Pembrolizumab ± platinum doublet chemotherapy	Tagged-amplicon sequencing of hotspots and coding regions from 36 genes	Rapid decrease in ctDNA post 21 days treatment prior to radiological assessment correlated with higher radiographic responses and long-term clinical outcomes	[30]
**Melanoma**	Ipilimumab (as anti-CTLA-4) and nivolumab (as anti-PD-1)	NGS	Simultaneous reduction of both tissue TMB and cTMB parameters after 3 weeks of starting treatment was able to identify responders vs. non-responders	[31]
**Metastatic gastric cancer**	Pembrolizumab	73-gene ctDNA NGS assay	Decreased ctDNA post treatment was associated with improved outcomes	[32,33]
**Advanced solid cancers**	ICIs	Targeted NGS panel	Higher quantity of ctDNA was associated with shorter time to failure and shorter OS compared to patients with lower quantity of ctDNA	[34]
**Neuroendocrine Cervical Carcinoma**	Nivolumab and Stereotactic Body Radiation Therapy (SBRT)	Deep sequencing	High cTMB correlated with observed high tTMB, as a consequence of a mismatch repair gene defect. This observation led to a therapeutic “match” with an anti-PD 1 antibody combined with radiation therapy, resulting in a durable response (10 + months).	[35]
**NSCLC**	Pembrolizumab±platinum/pemetrexed	Guardant360^®^ test, and MAGIC (Monitoring Advanced NSCLC through plasma Genotyping	Presence of TP53 mutations in ctDNA, confers poor survival both on ICI and chemotherapy. STK11 mutated patients (*n* = 9) showed worse OS (only if treated with ICIs). The presence of KRAS/STK11 co-mutation and KRAS/STK11/TP53 co-mutation affected OS only in patients treated with ICIs indicating a predictive role	[36]
**CTCs**				
**NSCLC**	Nivolumab	CTCs number, PD-L1+ CTCs	Contradictory result on the correlation of PD-L1 CTC with poor outcome	[37,38]
**Gastrointestinal tumours**	IBI308 (PD-1 monoclonal antibody)	PD-L1 expression	Responders have high expression of PDL1 on CTCs. After treatment: significant reduction of PD-L1+ CTCs and high-positive PD-L1 CTCs	[39]
**HNC**	Nivolumab	CTCs number, PD-L1+ CTCs	Post treatment CTC-positive patients had a shorter PFS, and PD-L1-positive CTCs were significantly associated with worse outcomes	[40]
**Melanoma**	Combinatorial immunotherapy (pembrolizumab, bevacizumab nivolumab/dabrafenib/trametinib)	CTC mRNA and DNA biomarker panels	Incorporation of a DNA biomarker with mRNA profiling increased overall CTC-detection capability. CTC-CTNNB1 was associated with progressive disease/stable disease compared to complete-responder patient status	[41]
**NSCLC,** **Hepatic carcinoma**	NK cell therapy	CTCs number	The decrease of CTC number is related to NK cell therapy efficacy	[42,43]
**AR-V7 positive metastatic PCa**	Ipilimumab and nivolumab	ARV7 expression, DNA-repair gene mutations and phenotypic heterogeneity	Higher mutations in DNA-repair-related genes in isolated CTCs and their phenotypic heterogeneity were associated with improved clinical outcomes in ARV-7 positive patients Presence of AR-V7+ CTCs was associated with CTCs heterogeneity, which correlated with the likelihood of a favourable response to immune-checkpoint inhibition	[44]
**Breast cancer**	Pembrolizumab	CTC mRNA	Patients with overexpression of PD-L1 have shown a better response to mono-immunotherapy	[45]
**NSCLC**	Pembrolizumab, nivolumab, atezolizumab	PD-L1 expression of CTCs	Upon disease progression, all patients demonstrated an increase in PD-L1+ CTCs, while no change or a decrease in PD-L1+ CTCs was observed in responding patients. An increase of PD-L1+ CTCs had the potential to predict resistance to PD-1/PD-L1 inhibitor	[46]
**Immune cells**				
**Metastatic melanoma**	Ipilimumab	T-cell repertoire (sequencing of the CD region 3 of the T-cell receptors)	TCR diversity was evaluated using a polymerase chain reaction assay, which measures TCR combinatorial diversity between V and J genes from genomic DNA. TCR repertoire TCR diversity was associated with clinical benefit but not overall survival	[47]
**NSCLC**	Pembrolizumab	The expression level of PD-L1 repertoire of T-cell were analysed using the PCR-based method)	Patients with low T-cell receptor diversity at baseline responded better to anti-PD-L1 therapy	[48]
**NSCLC**	Nivolumab	Frequency of immune-suppressive cells, including Tregs and MDSCs	The number of polymorphonuclear MDCSCs and TREG in peripheral blood was significantly higher in non-responders than in responders using flow cytometry	[49]
**Chronic lymphatic leukaemia**	Relatlimab	Lymphocyte activation gene 3 (LAG-3 expression level in T cells and NK cells)	The expression level of LAG-3 was analysed by in Silico mRNA analysis and serum level of LAG-3 by ELISA	[50]
**Relapsed Follicular Lymphoma**	Pidilizumab	The number of PB immune cells, the expression of the activating receptor NKG2D on NK cells	Increased number of PB immune cell observed using multiparametric flow cytometry. Whole genome gene expression profiling (GEP) was performed on core needle biopsies	[51]

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
