# Peer review of "Harnessing Liquid Biopsies to Guide Immune Checkpoint Inhibitor Therapy"

_cancers, 2022, doi:10.3390/cancers14071669_

Round 1
Reviewer 1 Report
Harnessing liquid biopsies to guide immune checkpoint inhibitor therapy.
This review is well written by authors to enhance understanding about liquid biopsies to improve immune checkpoint inhibitor therapy.
However, few points need to be addressed before considering approval for publication.
1. Please include all literature about liquid biopsies considering metastatic tumors. How does these liquid biopsies help in late-stage tumors?
2. Please include information on regulatory approval (FDA and EMA) of liquid biopsy in cancers.
Minor comments:
1. Some of the references are missing from the main texts. Please add relevant reference throughout this review article.
2. Please re-check all references and format accordingly as per authors guidelines of journal.
Author Response
Reviewer 1:
This review is well written by authors to enhance understanding about liquid biopsies to improve immune checkpoint inhibitor therapy.
However, few points need to be addressed before considering approval for publication.
Thanks for the overall positive assessment of our manuscript.
Please include all literature about liquid biopsies considering metastatic tumors. How does these liquid biopsies help in late-stage tumors?
We thank the reviewer for their comment but feel that while there is overwhelming literature regarding liquid biopsies being more informative for metastatic tumors, it is beyond the scope of this review to differentiate the use of liquid biopsies between stages of disease unless relevant for the decision to prescribe IO to the patients. We feel changes in the manuscript addressing this question is not warranted unless we misunderstood the exact issue the reviewer raises.
Please include information on regulatory approval (FDA and EMA) of liquid biopsy in cancers.
Thanks for pointing this issue out, we have now amended the text as follows and added the FDA approved liquid biopsy-based test for immunotherapy. Please check lines 75-81, 278-79, 511-513, and 579-582.Most FDA-approved companion diagnostics are still based on tumor tissue biomarkers for example FDA approved Roche (Ventana) PD-L1 IHC assay for tumor sections and pharmDx PD-L1 IHC.
Minor comments:
1. Some of the references are missing from the main texts. Please add relevant reference throughout this review article.
We have reinserted the references from our endnote library and added 20 more references to the text.
2. Please recheck all references and format accordingly as per authors’ guidelines of the journal.
The reference library was not correctly transferred when the text is imported which led to problems in references. We have reinserted every reference and old reference list has been deleted and is displayed in a numbered format.
Reviewer 2 Report
This is a very interesting review article on the role of liquid biopsy approaches as a valuable opportunity to predict and monitor immunotherapy response due to their logistic accessibility. This review can benefit from the following points.
- The authors write in the “Abstract” that “Liquid biopsy has the potential to circumvent tumor heterogeneity and to identify patients who may respond to IO…” , however in the following paragraphs they do not provide further explanations about the direct mechanisms underlying such a circumvent of tumor heterogeneity.
- The authors should provide detailed information on the role of extracellular vesicles as liquid biomarkers with potential use in immunotherapy response monitoring. In the paragraph above Figure 2, they mention that “ Tumor-derived entities present in bodily fluids may include circulating proteins, circulating tumor DNA (ctDNA) and RNA (ctRNA), extracellular vesicles (EVs) and circulating tumor cells (CTCs)” but in the following text they do not discuss the role of EVs as liquid biomarker.
- The role of microbiome and soluble PDL1 should be also briefly discussed.
Author Response
Reviewer 2:
This is a very interesting review article on the role of liquid biopsy approaches as a valuable opportunity to predict and monitor immunotherapy response due to their logistic accessibility. This review can benefit from the following points.
Thanks for the overall positive assessment of our manuscript.
The authors write in the “Abstract” that “Liquid biopsy has the potential to circumvent tumor heterogeneity and to identify patients who may respond to IO…” , however in the following paragraphs they do not provide further explanations about the direct mechanisms underlying such a circumvent of tumor heterogeneity.
We agree with the reviewer and the editions were made on Line 94-99 and 107-11.
The authors should provide detailed information on the role of extracellular vesicles as liquid biomarkers with potential use in immunotherapy response monitoring. In the paragraph above Figure 2, they mention that “ Tumor-derived entities present in bodily fluids may include circulating proteins, circulating tumor DNA (ctDNA) and RNA (ctRNA), extracellular vesicles (EVs) and circulating tumor cells (CTCs)” but in the following text they do not discuss the role of EVs as liquid biomarker.
A section defining the importance of exosomes and EVs in Immunotherapy response prediction has been added please see pg 13 lines 438-464 (Fig-2 has been amended accordingly as well).
The role of microbiome and soluble PDL1 should be also briefly discussed.
The role of Soluble PDL1 is added as an added section pg 13 lines 466-500 (Fig-2 has been amended accordingly as well).
Relevance of microbiome is mentioned in the future directions pg 15 line 591.